# Real-Time Detection and Short-Term Prediction of Blast Furnace Burden Level Based on Space-Time Fusion Features

**DOI:** 10.3390/s22145412

**Published:** 2022-07-20

**Authors:** Yanli Chen, Zhipeng Chen, Weihua Gui, Chunhua Yang

**Affiliations:** School of Automation, Central South University, Changsha 410083, China; 8207190918@csu.edu.cn (Y.C.); gwh@csu.edu.cn (W.G.); ychh@csu.edu.cn (C.Y.)

**Keywords:** blast furnace, burden level prediction and detection, mechanical stock rod, radar probe, space-time fusion features

## Abstract

Real-time, continuous and accurate blast furnace burden level information is of great significance for controlling the charging process, ensuring a smooth operation of a blast furnace, reducing energy consumption and emissions and improving blast furnace output. However, the burden level information measured by conventional mechanical stock rods and radar probes exhibit problems of weak anti-interference ability, large fluctuations in accuracy, poor stability and discontinuity. Therefore, a space-time fusion prediction and detection method of burden level based on a long-term focus memory network (LFMN) and an efficient structure self-tuning RBF neural network (ESST-RBFNN) is proposed. First, the space dimensional features are extracted by the space regression model based on radar data. Then, the LFMN is designed to predict the burden level and extract the time dimensional features. Finally, the ESST-RBFNN based on a proposed fast eigenvector space clustering algorithm (ESC) is constructed to obtain reliable and continuous burden level information with high accuracy. Both the simulation results and industrial verification indicate that the proposed method can provide real-time and continuous burden level information in real-time, which has great practical value for industrial production.

## 1. Introduction

The iron and steel industry is a major energy consumer and carbon emitter. As the most critical equipment for iron and steel smelting, blast furnaces account for about 60%~70% of the comprehensive energy consumption and cost of iron and steel [1], and CO_2_ emissions account for more than 90% [2]. Blast furnace burden level is a key parameter of blast furnace production [3]. Real-time and continuous blast furnace burden level with high accuracy is of great significance for controlling the charging process, ensuring smooth operation of blast furnace [4], reducing energy consumption and emissions and improving the blast furnace output [5]. With the increasing requirements for refinement and intelligence of the smelting process, real-time continuous high-precision measurement of blast furnace burden levels is becoming increasingly important [6].

A blast furnace is an enclosed large countercurrent reactor [7], which has high temperatures, high pressure and much dust, and its environment is extremely harsh [8]. At present, the detection methods for the burden level of blast furnaces are mainly divided into contact methods and non-contact methods [9]. The mechanical stock rod (mechanical probe) is the main contact detection device, which is composed of a steel wire rope, a heavy hammer, a winch motor outside the furnace, a reducer, an encoder and a master controller for detection control [10]. This detection device has a simple structure and reliable mechanical transmission mechanism and can adapt to the complex and harsh environment in the blast furnace. Therefore, the burden level information obtained by a mechanical probe has high accuracy [11]. However, it can neither achieve continuous measurement to ensure the continuity of burden level information nor track the change in burden level in time when the blast furnace shows slip, inspect, or collapse conditions [12]. A radar probe is the most widely used non-contact detection device, which consists of an antenna transmitter, a receiver, and a display screen [13]. The electromagnetic wave signal is firstly transmitted to the burden surface, and then it is reflected by the burden surface and received by the receiver to form a radar echo signal to measure the burden level. Its working principle determines that the radar probe can obtain real-time and continuous burden level information. However, due to the high-speed gas flow and high concentration of dust in the furnace, it is easy to interfere with the echo signal of the radar probe, which distorts the measurement data and reduces the measurement accuracy and reliability [14]. Therefore, obtaining high-precision and real-time continuous measurement data of the burden level is challenging to solve.

Z.H. Ma et al. proposed a closed-loop speed control method, which realizes the anti-overturning control in the variable torque control by changing the anti-overturning torque, so as to make the mechanical probe drop at a uniform speed and ensure its stable operation. On the basis of the mechanical probe tracking process, the descending speed of the burden surface is used to judge whether an abnormal condition occurs [15]. Based on the mechanical probe data, J. Hinnel and H. Saxén established the relationship between the burden thickness and the detection data by using the neural network based on mechanical probe data and systematically analyzed the falling rate of the furnace burden [16]. Although this research has improved the accuracy of measuring the burden level with a mechanical probe, it still cannot overcome the problem of discontinuity in the measuring process. For the research of measuring burden level with a radar probe, in order to solve the problem of low accuracy and reliability of a radar probe, S. Schuster et al. used an integrated millimeter-wave design circuit and advanced signal processing technology to obtain the average burden level information of a blast furnace in real-time [17]. Dominik Zankl et al. proposed a BLASTDAR radar sensing system for real-time radar imaging of blast furnace burden surface [18]. X.Z. Chen et al. proposed a new forward-looking staring high-resolution imaging method for synthetic aperture radar (SAR) to further improve the definition and resolution of radar imaging [19]. The above methods have made a breakthrough in the real-time measurement of burden level with radar probe, especially for the radar imaging of the whole burden level, which is of great significance to blast furnace regulation and smelting. However, there is still a lack of efficient and feasible technical methods to overcome the defects of low measurement accuracy and large fluctuations of radar probes. In recent years, with the rapid development of deep learning technology, X.P. Liu et al. proposed a new codec structure composed of a convolutional neural network and long-term and a short-term memory network to classify distorted signals and perform regression in a learning manner to model long-term historical measurements [5]. Compared with the traditional manual denoising method, this method saves time and energy, but its accuracy still requires improvements. Q.D. Shi et al. combined high-temperature metallurgy, radar detection and image processing, and a new blast furnace surface deep learning detection method of a blast furnace smelting state visualization system for a burden surface based on energy weight is proposed, which realizes the visualization of blast furnace smelting state and digitization of burden surface information [20]. H. Wang et al. proposed a key point estimation method based on learning combined with a key point-based connected region noise reduction algorithm (KP-CRNR) to reconstruct the key points in the BSP image measured by the radar probe. This method improves the measurement accuracy of the radar probe from the perspective of the working principle of the sensor [21]. From the above research, the introduction of deep learning technology improves the accuracy of radar probes for burden level measurement to a certain extent. However, most of the existing studies ignore the important significance of a mechanical probe for burden level measurement [22]. It is difficult to fundamentally solve the current issue of low accuracy and large fluctuation of radar probe level measurement data just from the characteristics of radar data.

Therefore, the blast furnace smelting burden distribution mechanism is further analyzed, and it is discovered that the burden level has the features of slowly changing in the time dimension and rapidly changing in the space dimension. That is, the burden level fluctuates rapidly in the space dimension due to the rapid change in smelting conditions and environment in the furnace, while its periodic features, alternating frequency and other time dimensional features do not follow the rapid change in dimensional space features. Based on this, a prediction and detection method of burden level is proposed based on a long-term focus memory network and an efficient structure self-tuning RBF neural network. Figure 1 shows the research framework. Firstly, the space feature regression model of radar data based on mechanical probe data is established to extract the space dimensional features of blast furnace burden level. Then, the mechanical stock rod data are set as the standard long-term memory, and a long-term focus memory depth network structure is constructed. Combining the historical periodic radar data and the mechanical probe data, not only the burden level information of the next cycle is predicted with high precision but also the time dimensional features of which are accurately extracted. Finally, based on a proposed fast eigenvector space clustering algorithm, an efficient structure self-tuning RBF neural network is constructed. By integrating the advantages of mechanical probe data and radar data, the real-time continuous detection of burden level information with high precision is realized. Simulation results and industrial data verification demonstrate that the proposed method for blast furnace burden level prediction and continuity-accuracy detection has good real-time performance, high precision and obvious industrial practical value.

The remainder of the paper is organized as follows. In Section 2, the space dimensional feature extraction based on the space feature regression model of radar data is introduced. In Section 3, the long-term focus memory network is developed to realize the burden level prediction and tim-dimensional feature extraction. In Section 4, the efficient structure self-tuning RBF neural network based on a fast eigenvector space clustering algorithm is presented to realize real-time, continuous and accurate burden level detection. The simulation and industrial verification results are given in Section 5. Section 6 gives the discussion and the conclusions.

## 2. Space Dimensional Feature Extraction

### 2.1. Analysis of the Mechanical Probe Data and Radar Data

The distribution mechanism of a blast furnace is batch charging, which has the nature of periodicity. A distribution cycle of a blast furnace is 3 to 6 min. Generally, the first 1.5 min is the feeding period in a cycle, and the last 1.5 to 6 min is the idle period in a cycle. As shown in Figure 2, the change in blast furnace burden level has the periodic subsection nonlinear characteristic. At the same time, it can be seen that in a distribution cycle, the burden level decreases with the increase in burden surfaces in the feeding period and increases with the decrease in burden surfaces in the idle period. Analyzing the mechanical probe data and radar data indicates that the burden level data sampled by the mechanical probe is discrete, and the mechanical probe is randomly sampled once in each distribution cycle, while the burden level data sampled by the radar probe is continuous. The different sampling properties of the two kinds of probe data make it difficult for the two detection data to match accurately within an infinite time distribution, so it is difficult to integrate the advantages of the two detection data, which makes it very challenging to obtain the blast furnace burden level in real-time with high accuracy. In order to accurately match the data of the two probes and build a strong correlation between the two detection data, a sliding window with a width of N is used to construct a single mapping relationship between the data of the radar probe and the mechanical probe, and Figure 3 shows the schematic of the data processing process of the sliding window. In this way, the matching problem of the two infinite dimensional detection data is transformed into a corresponding problem of the two detection data, providing a data foundation for the subsequent algorithm in this paper. According to the previous research, when the window size is N=10, the accuracy of burden level detection is the highest [22].

### 2.2. Space Feature Regression Model of the Radar Data

The change in the burden level has the space features of material falling speed, acceleration, and jerk, etc. Since the radar data can fully reflect the space change in the burden level, based on the mechanical probe data, a piecewise space feature regression model is established for the radar data to obtain the space feature parameters of blast furnace burden level change. We choose the fifth order curve to fit it, with each coefficient order corresponding to some physical significance. The time sequence of the radar data in a single period is set as $Q_{i}=\{(t_{i1},l_{i1}),(t_{i2},l_{i2}),\cdots (t_{im},l_{im})\}$. Piecewise nonlinear regression fitting of the radar data by the fifth-order polynomial is described as
(1)$$l_{i}=\left\{\begin{array}{l} s_{di}+v_{di}t+a_{di}t^{2}+j_{di}t^{3}+h_{di}t^{4}+k_{di}t^{5}+\varepsilon_{i},\;t_{\max i}\leq t\leq t_{\min i}\\ s_{ui}+v_{ui}t+a_{ui}t^{2}+j_{ui}t^{3}+h_{ui}t^{4}+k_{ui}t^{5}+\varepsilon_{i},\;t_{\min i}\leq t\leq t_{\max i+1} \end{array}\right.$$
where i=1,2,⋯,n presents the i th cycle; u and d present the feeding period and the idle period, respectively; s,v,a,j,h and k are the initial position, falling speed, acceleration, jerk, jerk changing rate and high-order term of the burden level, respectively; $t_{\max i}$ and $t_{\max i+1}$ are the beginning time and the ending time of the cycle; $t_{\min i}$ is the dividing time between the feeding period and the idle period of this cycle; $\varepsilon_{i}$ is the measurement error.

To improve the accuracy and reliability of the data, combining the measurement data sequence of the mechanical probe $\left(t_{Ri},l_{Ri}\right)$ as the standard reference point into Equation (1), the space feature regression model of the radar data based on the mechanical probe data is described as
(2)$$l_{i}=\left\{\begin{array}{l} s_{di}+v_{di}t+a_{di}t^{2}+j_{di}t^{3}+h_{di}t^{4}+k_{di}t^{5}+\varepsilon_{i}+\left(L_{R}-l_{Ri}\right),\;t^{'}{}_{\max i}\leq t\leq t_{\min i+1}\\ s_{ui}+v_{ui}t+a_{ui}t^{2}+j_{ui}t^{3}+h_{ui}t^{4}+k_{ui}t^{5}+\varepsilon_{i}+\left(L_{R}-l_{Ri}\right),\;t_{\min i+1}\leq t\leq t^{'}{}_{\max i+1} \end{array}\right.$$
where $L_{R}$ is the radar data closest to the detection time $t=t_{i}$ of the mechanical probe in the i th cycle. To ensure that the continuous regression curve of radar data segments is continuous at the periodic junction, the dividing points of the cycle $t_{\max i}$ and $t_{\max i+1}$ divided based on discrete data are adjusted to $t^{'}{}_{\max i}$ and $t^{'}{}_{\max i+1}$. Setting $l_{di}\left(t\right)$ and $l_{ui}\left(t\right)$ as the burden level value in the falling period and rising period of the i th cycle, for $l_{di}\left(t_{\max i}^{'}\right)$ in the feeding period, $l_{di}\left(t_{\max i}^{'}\right)=l_{u_{i-1}}\left(t_{\max i+1}^{'}\right)$, and for $l_{ui}\left(t_{\max i+1}^{'}\right)$ in the idle period, $l_{ui}\left(t_{\max i+1}^{'}\right)=l_{d_{i+1}}\left(t_{\max i}^{'}\right)$. Figure 4 shows the original burden level measurement curve obtained based on the space feature regression model of radar data and the piecewise regression nonlinear (PRNN) curve after fine-tuning the periodic dividing points. The area between the red solid lines in the figure represents the i th cycle, and the area between the red dotted lines represents the idle period of the i th cycle and the feeding period of the i+1 th cycle. Compared with the original radar data curve, the space feature regression curve of radar data is smoother and more coherent.

## 3. Time Dimensional Feature Extraction and Burden Level Prediction Based on Long-Term Focus Memory Network

The time dimensional change in blast furnace burden level corresponds to its space change features, and it has the time dimensional features of extreme time points such as speed, acceleration, jerk and jerk changing rate. Considering the complex and changeable environment in the blast furnace, the radar data are different in different environments. However, in the adjacent cycle, due to the extremely short sampling time interval and the small change in the detection environment, the feature changes in the radar detection data tend to be consistent in the time dimension. According to the distribution mechanism of the blast furnace, the distribution of the blast furnace does not change significantly in a short time; that is, the periodic feature in the time dimension does not change, so the slow variation feature of radar data reflects the features that the distribution mechanism does not change in a short time, and it can well predict the change trend of the burden level. Therefore, based on the slow variation feature of radar data in the time dimension, the LFMN deep network structure is proposed. By training the LFMN network, the radar data of the i+1 th cycle is predicted. The biggest advantage of this deep network in burden level prediction is that it introduces the attention mechanism with the mechanical probe data as accurate and reliable, slowly varying burden level information, which improves the reliability and accuracy of burden level information predicted based on the slow variation features of radar data. According to the actual distribution mechanism of the blast furnace, we combine the radar data of the i−1 th and the i th cycle with the predicted radar data of the i+1 th cycle, and the time dimensional features of the i+1 can be extracted.

The LFMN network is composed of an input layer, a long-term focus memory layer consisting of multiple memory units, and an output layer. Its structure is shown in Figure 5. The input samples of the network input layer are set as follows,
(3)$$x_{i}(k)=\left\{\begin{array}{l} \left(t^{'}{}_{\max i},t_{\min i+1},s_{di},v_{di},a_{di},\right.\left.j_{di},\gamma_{i},T_{i},f_{i},l_{k-N+1}^{i},\cdot \cdot \cdot ,l_{k}^{i}\right),t_{\max i}^{'}<t_{k}^{i}\leq t_{\min i+1}\\ \left(t_{\min i+1},t^{'}{}_{\max i+1},s_{ui},v_{ui},a_{ui}\right.,\left.j_{ui},\gamma_{i},T_{i},f_{i},l_{k-N+1}^{i},\cdot \cdot \cdot ,l_{k}^{i}\right),t_{\min+1}<t_{k}^{i}\leq t_{\max i+1}^{'} \end{array}\right.$$
where i represents the cycle of the current input data, i≤q, and q represent the last one of the detected cycles. For the i th cycle, $\left(t_{\max i}^{'},t_{\min i+1}\right)$ is the feeding period, $\left(t_{\min i+1},t_{\max i+1}^{'}\right)$ is the idle period, $\left(\gamma_{i},T_{i},f_{i}\right)$ are the working condition parameters that have the greatest correlation with the blast furnace burden level, representing the opening of the material flow valve, the furnace top temperature and the furnace top air volume, respectively. $\left(l_{k-N+1}^{i},\cdots ,l_{k}^{i}\right)$ is the N radar data in the k th window of the i th cycle processed by the sliding window, and the sliding windows are shown in Figure 3. The output of the network is set as the predicted n radar data of the q+1 th cycle $\left(l_{1}^{q+1},\cdots ,l_{n}^{q+1}\right)$.

For each memory unit in the long-term focus memory layer, it contains two state variables h and C, saving the short-term memory and long-term focus memory of the network. At time t, each memory unit has three inputs, which are the input sample $x_{t}$ of the time t, the short-term memory $h_{t-1}$ of the time t−1, and the long-term focus memory $C_{t-1}$. The internal structure of each long-term focus memory unit is shown in the right figure of Figure 4, including four control gates, which are the input gate, forgetting gate, guiding gate, and output gate. The long-term focus memory C is controlled by the input gate and the forgetting gate, which determines how many input states at the current time t are updated into the long-term focus memory.

The forgetting gate determines how much of the long-term focus memory $C_{t-1}$ of the time t−1 is retained for the current time t. Therefore, the long-term focus memory $C_{t}$ of the time t can be described as
(4)$$\begin{array}{l} C_{t}=f_{t}*C_{t-1}+i_{t}*\tilde{C_{t}}\\ \left\{\begin{array}{l} f_{t}=\sigma \left(w_{f}\cdot \left[h_{t-1},x_{i}\left(k\right)\right]+b_{f}\right)\\ i_{t}=\sigma \left(w_{i}\cdot \left[h_{t-1},x_{i}\left(k\right)\right]+b_{i}\right)\\ {\tilde{C}}_{t}=\tanh\left(w_{c}\cdot \left[h_{t-1},x_{i}\left(k\right)\right]+b_{c}\right) \end{array}\right. \end{array}$$
where $f_{t},i_{t}$ represent the forgetting gate and the input gate; $w_{f},w_{i}$ are the weight matrix of forgetting gate and input gate in network training; $b_{f},b_{i}$ are the bias of the forgetting gate and input gate; σ represents the sigmoid activation function; $\tilde{C_{t}}$ is the long-term memory of the current time t; $w_{c},b_{c}$ represent the weight matrix and the bias of the current inputting long-term state.

The guiding gate is set to retain the focus information of the long-term focus memory unit. This is because the mechanical probe data contains the most important long-term changing information of the burden level, and it has the advantages of high accuracy and reliability. The model of the guiding gate is described as
(5)$$\left\{\begin{array}{l} C_{t}{}^{'}=\theta C_{y}+\left(1-\theta \right)C_{t}\\ C_{y}=\sum_{j=1}^{i}\left(f_{j}+\left(L_{Rj}-l_{Rj}\right)\right) \end{array}\right.$$
where $C_{t}^{'}$ represents the long-term focus memory of the time t stored in the long-term focus memory layer; $C_{y}$ represents the long-term burden level focus information, which combines radar data and mechanical probe data; $L_{Rj}$ is the radar data closest to the mechanical probe data in the j th cycle. $l_{Rj}$ is the mechanical probe data in the j th cycle; $f_{j}$ represents the burden level prediction model based on radar data of the j th cycle, which can be expressed as
(6)$$\left\{\begin{array}{l} f_{j}=\sum_{r=1}^{5}\left(\alpha_{jr}*\exp\left(-{\left(\left(t-b_{jr}\right)/c_{jr}\right)}^{2}\right)\right)\\ \left[\begin{array}{l} a_{jr}\\ b_{jr}\\ c_{jr} \end{array}\right]=\left[\begin{array}{ccccc} a_{j1} & a_{j2} & a_{j3} & a_{j4} & a_{j5}\\ b_{j1} & b_{j2} & b_{j3} & b_{j4} & b_{j5}\\ c_{j1} & c_{j2} & c_{j3} & c_{j4} & c_{j5} \end{array}\right] \end{array}\right.$$
where $\alpha_{jr},b_{jr},c_{jr},r=1,2,\cdots ,5$ are the radar model’s fifth order Gaussian fitting coefficients of the j th cycle of the radar data.

The output gate is responsible for controlling the contribution of long-term focus memory to the network output value at the current time t. Its effect can be quantitatively described as
(7)$$\left\{\begin{array}{l} h_{t}=o_{t}*\tanh\left(C_{t}{}^{'}\right)\\ o_{t}=\sigma \left(w_{o}\cdot \left[h_{t-1},x_{i}\left(k\right)\right]+b_{o}\right) \end{array}\right.$$
where $o_{t}$ represents the output gate, $w_{o},b_{o}$ are the weight matrix and the bias of output gate. The LFMN network can be trained by the error back propagation algorithm, and the predicted n burden level values of the i+1 th cycle are
(8)$$h_{i+1}=\left(l_{1}^{i+1},\cdot \cdot \cdot ,l_{n}^{i+1}\right)$$

Combining the radar data of the i−1 th and the i th cycle with the predicted radar data of the i+1 th cycle, the time dimension features of the i+1 can be extracted through the following Equation (9)
(9)$$\left\{\begin{array}{l} l_{k}^{i}{}^{\prime }=\frac{\alpha l_{k}^{i-1}+\beta l_{k}^{i}+\nu l_{k}^{i+1}}{3}\\ \alpha +\beta +\nu =1\;\;\;,\;\;\;k=1,2,\cdot \cdot \cdot ,n \end{array}\right.$$
where k represents the predicted burden level information of the k th cycle, and α,β,ν are the data fusion weight of the i−1,i,i+1 th cycle, respectively. The weights are selected based on experience, which are set as $\left(0.25,0.25,0.5\right)$ in this paper. Then, for the burden level sequence $l_{k}^{i}{}^{'}$ of the i th cycle with time dimensional features of multiple cycles, the space feature regression model of radar data is used for regression to obtain the continuous function $l_{i}^{'}$ of the burden level change in the cycle, and the time dimensional features of blast furnace burden level of the i th cycle can be extracted, as shown below
(10)$$\left\{\begin{array}{l} c_{tid}=\left(t_{idv\min},t_{idv\max},t_{ida\min},t_{ida\max},t_{idj\min},t_{idj\max}\right)\\ c_{tiu}=\left(t_{iuv\min},t_{iuv\max},t_{iua\min},t_{iua\max},t_{iuj\min},t_{iuj\max}\right) \end{array}\right.$$
where $t_{idv\min},t_{idv\max},t_{ida\min},t_{ida\max},t_{idj\min},t_{idj\max}$ and $t_{iuv\min},t_{iuv\max},t_{iua\min},t_{iua\max},t_{iuj\min},t_{iuj\max}$ are the time points at which the extreme values of burden level speed, acceleration and jerk are taken in the feeding period and the idle period of the i th cycle.

## 4. ESST-RBFNN

Based on the slow change features in time dimension and fast change features in the space dimension of the burden level, integrating the advantages of the two detection data and overcoming the common shortcomings, the efficient structure self-tuning RBF neural network (ESST-RBFNN) is designed, and the real-time continuous detection of blast furnace burden level with high accuracy can be realized. The structure of the network will change with the change in input, which adapts to different blast furnace environments and have industrial universality. At the same time, an efficient eigenvector space clustering (ESC) algorithm is proposed, which improves the efficiency of training and ensures the real-time detection of the burden level. Firstly, the input samples of the algorithm are composed of the time dimensional features and space dimensional features of the burden level, the radar data processed by the sliding window and the key working condition parameters. Secondly, based on the proposed fast eigenvector space clustering algorithm, the clustering center and width of the input samples are determined. Thirdly, based on the RBF neural network, a dynamic self-tuning network structure is designed. According to the characteristics of input samples, the number of hidden nodes is automatically increased and clipped to realize that the network structure changes with the characteristics of input samples, and the relevant parameters of the network structure are adjusted and updated independently to realize the self-tuning of the structure and complete the learning and training process of the ESST-RBFNN algorithm. Finally, the trained network is used to output the burden level information at the next time, and the whole algorithm is realized.

### 4.1. Fast Eigenvector Space Clustering Algorithm (ESC)

The input sample consists of four parts, which are the key operating condition parameters, space dimensional features, time dimensional features of each cycle and the radar data processed by the sliding window in this cycle. Its specific form is shown as follows
(11)xi(k)=t'maxi,tmini+1,ctid,sdi,vdi,adi,jdi,γi,Ti,fi,lk−N+1i,⋅⋅⋅,lki,                                                                     tmaxi'<tki≤tmini+1tmini+1,t'maxi+1,ctiu,sui,vui,aui,jui,γi,Ti,fi,lk−N+1i,⋅⋅⋅,lki,                                                                     tmin+1<tki≤tmaxi+1'
where xik represents the input sample of the k th window in the i th cycle. It can be analyzed that radar data are the most critical data in the input samples. In a single cycle, burden level changes are usually up and down the standard burden level, which leads to the typical accumulation of radar data. In addition, the environment in the furnace is bad, meaning that the echo-based radar data fluctuate greatly, so the accuracy is not high, and the signal-to-noise ratio is low. To overcome the interference of the furnace environment, the fast eigenvector space clustering algorithm (ESC) is proposed to reveal the intrinsic correlation of the input data. The algorithm clusters n input sample datasets X^=x1,x2,⋯xk,⋯xn into K categories. The clustering center vector set is C=c1,c2,⋯cK, and the input sample data set is centralized as X=(x1,x2,…,xn)T, where xi∈ℝd, i=1,2,…,n. In the d-dimensional data space, the problem of making the clustering center vector describe the internal clustering characteristics of input samples as much as possible can be equivalent to the optimization problem, as shown in the following Equation (12)
(12)minJK=∑inxi2−∑k=1K1nk∑Xi,cK∈CxicKT
where nK is the number of input samples in category K. Set the original indicator vector matrix of the category feature space as GK=(g1,g2,…,gK)T,where each row of the original indicator vector matrix of the category feature space has only one non-zero value, and each column can have multiple non-zero values, which indicates that each datum only belongs to the category represented by the column and does not belong to other categories. The matrix HK=(h1,h2,…,hK)T is obtained by the rotation equivalent transformation of the matrix GK, where hK=(0,…,0,1,…,1∵nK,0,…,0)T/nK, and the non-zero value in hK represents the corresponding sample belonging to the category K in the category feature space. After introducing the feature space indicator vector, the optimization objective function can be converted into the following Equation (13).
(13)minJK=∑inxi2−Tr(HKXXTHKT)

It should be noted that among the K clustering categories in the sample space X, the mean center of the sample has an impact on the embodiment of the intrinsic clustering characteristics of the samples. In order to eliminate this impact, the input sample space is centered into $\tilde{X}={\left({\tilde{x}}_{1},{\tilde{x}}_{2},\ldots ,{\tilde{x}}_{n}\right)}^{T}$, where ${\tilde{x}}_{i}=x_{i}-\bar{x}$ is the centralized data matrix obtained by removing the average value of the given data set $\bar{x}=\sum_{i}^{n}x_{i}/n$. The optimization objective function is rewritten as the following Equation (14)
(14)$$\min J_{K}=\sum_{i}^{n}{\left({\tilde{x}}_{i}+\bar{x}\right)}^{2}-Tr\left(H_{K}\left(\tilde{X}+\bar{X}\right){\left(\tilde{X}+\tilde{X}\right)}^{T}H_{K}^{T}\right)$$

To further simplify the optimization objective function, a K×K orthogonal transformation matrix T is constructed, and the last column of the matrix is set as
(15)$$T_{K}={\left(\sqrt{n_{1}/n},\sqrt{n_{2}/n},\ldots ,\sqrt{n_{K}/n}\right)}^{T}$$

Then the transformation matrix is linearly transformed into the indicator vector matrix of the category feature space $H_{K}$, and the transformed matrix is $Q_{K}$,
(16)$$Q_{K}=H_{K}^{T}T$$

The last column of the matrix $Q_{K}$ is given as
(17)$$\begin{array}{l} q_{k}={\left(\sqrt{n_{1}/n}h_{1}+\sqrt{n_{2}/n}h_{2}+\ldots +\sqrt{n_{K}/n}h_{K}\right)}^{T}\\ \;\;\;\;\;\;=\sqrt{1/n}e^{T}\;\;,\;\;\;\;\;\;e^{T}={\left(1,\ldots ,1,\ldots ,1\right)}^{T} \end{array}$$

By analyzing the relevant properties of the indicator vector matrix $H_{K}$ in the feature space, it is found that its vector $h_{K}$ has the property of pairwise orthogonality, and it can be expressed in the following form,
(18)$$\left\{\begin{array}{l} h_{{}_{K}}^{T}h_{L}=\delta_{KL}\\ \delta_{KL}\;\;\;\;\;=\;\;\;1\;\;\;\;,\;\;\;\;K=L\\ \delta_{KL}\;\;\;\;\;\;=\;\;0\;\;\;,\;\;\;\;otherwise \end{array}\right.$$

For the matrix $Q_{K}$, the following Equation (19) holds
(19)$$q_{K}^{T}q_{L}=T_{K}^{T}h_{{}_{K}}^{T}h_{L}T_{L}=\delta_{KL}$$

Set $Q_{K-1}=(q_{1},q_{2},\ldots ,q_{K-1})$, then using Equation (19) derives the following Equation (20)
(20)$$Q_{K-1}^{T}Q_{K-1}=I_{K-1}$$

Substituting Equations (16) and (17) together, the rewritten matrix $Q_{K}=[Q_{K-1},q_{K}]$ into the optimization objective Equation (14) for a series of derivation, transformation and simplification, the following Equation (21) can be obtained,
(21)$$J_{K}=Tr\left(\tilde{X}{\tilde{X}}^{T}\right)-Tr\left(Q_{K-1}^{T}\tilde{X}{\tilde{X}}^{T}Q_{K-1}\right)$$

Considering that $Tr\left(\tilde{X}{\tilde{X}}^{T}\right)$ is a constant larger than zero and it is determined by the input sample space, so minimizing $J_{K}$ and maximizing $J_{D}=Tr\left(Q_{K-1}^{T}\tilde{X}{\tilde{X}}^{T}Q_{K-1}\right)$ is equivalently.

Using KyFan theorem, for a real symmetric matrix $A=\tilde{X}{\tilde{X}}^{T}$ with eigenvalues, $\lambda_{1}\geq \lambda_{2}\geq \cdots \lambda_{n}$ are its eigenvalues and $\left(\nu_{1},\nu_{2},\cdots ,v_{n}\right)$ are its eigenvectors. Under the constraint of $Q_{K-1}{}^{T}Q_{K-1}=I_{K-1}$, the maximum value of $J_{D}$ is $Q_{K-1}=\left(\nu_{1},\nu_{2},\cdots ,\nu_{K-1}\right)U$, where U is a K×K orthogonal matrix, and the maximum value is
(22)$$maxJ_{D}=\lambda_{1}+\lambda_{2}+\cdots +\lambda_{K-1}$$

In combination with Equation (21), the upper and lower boundary of the optimization objective can be obtained after proper and reasonable expansion as
(23)$$n\bar{{\tilde{X}}^{2}}-\sum_{k=1}^{K}\lambda_{k}<J_{K}<n\bar{{\tilde{X}}^{2}}$$
where $n\bar{{\tilde{X}}^{2}}$ is the total variance of samples X.

Because the indicator vector matrices $H_{K-1}$ and $Q_{K-1}$ of the category feature space have the following relationship
(24)$$Q_{K-1}=H_{K-1}^{T}T_{K-1}$$
where $T_{K-1}$ is a $\left(K-1\right)\times \left(K-1\right)$ orthogonal matrix. Then the indicator vector matrix K $H_{K-1}$ of the category feature space can be
(25)$$H_{K-1}^{T}=\left(\nu_{1},\nu_{2},\cdots ,v_{K-1}\right)$$

In this way, the optimization objective function is guaranteed to obtain the minimum value. Then, according to the indicator vector matrix $H_{K-1}$, for any one of the former K−1 clustering categories in which the input samples X are divided into K categories, the corresponding indicator vector of category feature space is $H_{j}=\left(v_{j1},v_{j2},\cdots ,v_{jn}\right)$, which can be used to judge whether the n samples belong to the category j. The discriminant is as follows,
(26)$$I\left(C_{j}\right)={\left\{x_{i}\left|v_{ji}>0\right.\right\}}_{i=1,2,\cdots ,n\;\;\;\;j=1,2,\cdots K-1}\;\;$$
where $I\left(C_{j}\right)$ represents the input sample set belonging to category j. Note that the distribution of the former K−1 categories of the input sample set $\hat{X}$ can be determined by using Equation (26). For the category, it is common practice to classify the samples that do not belong to the former K−1 categories in the input sample set $\hat{X}$ as the category K. The mathematical description is as the following
(27)$$I\left(C_{K}\right)=\hat{X}-\sum_{j=1}^{K-1}{}^{\cup }I\left(C_{j}\right)$$

For each clustering center vector, it can be calculated by the following Equation (28)
(28)$$c_{j}={\sum_{x_{s}\in I(C_{j})}x_{s}/m}_{j=1,2,\cdots ,K}$$
where m is the elements number of the set $I\left(C_{j}\right)$.

### 4.2. Efficient Structure Self-Tuning RBF Neural Network (ESST-RBFNN)

Due to the strong nonlinear correlation between the radar data and the mechanical probe data, in order to effectively integrate the advantages of the two-detection data and improve the burden level measurement accuracy, a network named ESST-RBFNN with a flexible and self-tuning structure is presented. The network structure is shown in Figure 6, and the mathematical model of the network is described as
(29)$$\left\{{}_{g_{j}\left(x_{i}\left(k\right)\right)=\exp\left(-\frac{{\left\|x-c_{j}\right\|}^{2}}{\sigma_{j}{}^{2}}\right)\;,\;j=1,2,\ldots ,K}^{\begin{array}{l} f\left(x_{i}\left(k\right)\right)=y_{i}\left(k\right)=\sum_{j=1}^{m}w_{j}g_{j}\left(x_{i}\left(k\right)\right) \end{array}}\right.$$
where $x_{i}\left(k\right)$ is the k th fusion input sample of the i th cycle; $y_{i}\left(k\right)$ is the k th output burden level data of the i th cycle; $g_{j}\left(x_{i}\left(k\right)\right)$ is gaussian basis excitation function of the j th hidden node; $c_{j}$ is the center of the j th basis function; $\sigma_{j}$ is the width of the j th basis function; K is the number of hidden nodes; $w_{j}$ is the output weight. The number of hidden nodes, the center and the width of the basis function can be determined by the clustering results of the ESC algorithm for the input samples set $\hat{X}$. Generally, the number of hidden nodes is the number of clustering classes, and the centers of the basis functions are the clustering centers. The width of the basis function $\sigma_{j}$ is determined by the following Equation (30)
(30)$$\left\{{}_{\begin{array}{l} \sigma_{j}=\frac{2d_{\max}^{j}}{3}\;\;\;\;\;\;\;(1\leq j\leq K) \end{array}}^{\begin{array}{l} G_{i}=\left\{x_{i}\left(k\right)\in I(C_{j})\right\}\\ d_{\max}^{j}=\underset{x_{i}\left(k\right)\in G_{i}}{\max}\left\{{\left\|x_{i}\left(k\right)-c_{j}\right\|}^{2}\right\} \end{array}}\right.$$
where $d_{\max}^{j}$ is the maximum distance from the clustering center to the sample point.

To determine the weight $w_{j}$, the output matrix of the hidden layer H can be calculated according to Equation (29), which is a n×K matrix. The mechanical probe data corresponding to the input samples used for neural network training can be set as a matrix Y, and the network weight vector W can be obtained by solving the generalized inverse as the following
(31)$$W={[H^{T}H]}^{-1}H^{T}Y$$

According to the network structure shown in Figure 6, the network has the characteristic of dynamic structure self-tuning, which can automatically increase the number of hidden layer nodes, change the network structure and tune the network parameters itself with the increase in training samples. The main steps of ESST-RBFNN are as follows,

**Step 1:** According to the characteristics of input samples, set the value of category discrimination radius b as a positive real number.

**Step 2:** The algorithm begins. When the radar data of the first sliding window of the first cycle is sampled, the first input sample is constructed as $x_{1}^{1}$ according to Equation (11), and the first cluster is formed and K=1, the clustering center $C=\left\{c_{1}\right\}$, $c_{1}=x_{1}^{1}$, and the first hidden node of the network hidden layer $h_{1}$ is generated.

**Step 3:** When the k th input sample data of the i th cycle is sampled, the number of samples n=k, k=2,3,⋯. There have been D−1 clustering categories. The number of classes is K=D−1. The clustering centers C=c1,c2,⋯cD−1, and the hidden nodes are hp, p=1,2,3,⋯,K. The input sample data matrix is X=xi1,xi2,⋯,xinT. Set ς=(xis,yis)xis∈ℝd,yis∈ℝl,s=1,2,…,k−1 as the network training set, where ℝd and ℝl are the training input sample set and its corresponding mechanical probe data set. If minxik−cjcj∈C2<b, keep the current clustering result and the network structure unchanged, and go to Step 7. Otherwise, dynamically adjust the network structure, increase the number of the hidden layer nodes by 1, go to Step 4, update the clustering result, and tune the network structure itself.

**Step 4:** Centralizing the input sample data matrix X into $\tilde{X}={\left({\tilde{X}}_{1},{\tilde{X}}_{2},\ldots ,{\tilde{X}}_{n}\right)}^{T}$, and calculate the former D eigenvectors $v_{1},\ldots ,v_{D}$ of $\tilde{X}{\tilde{X}}^{T}$ to obtain the feature space indicator vector of the category to which each sample belongs. All samples are reclassified by using Equations (26) and (27) to obtain a new set of categories $I=\left\{I\left(C_{1}\right),I\left(C_{2}\right),\cdots ,I\left(C_{D}\right)\right\}$.

**Step 5:** Use Equation (28) to calculate and update the clustering center vector set and combine Equation (30) to synchronously update the center and width of the basis function of the hidden nodes in the hidden layer of the network.

**Step 6:** Use Equation (31) to calculate and update the network weight vector to complete the establishment and learning of ESST-RBFNN.

**Step 7:** Use Equation (29), set $x_{i}^{k}$ as the model input, and the current burden level information $y_{i}^{k}$ can be obtained through real-time calculation. With the loop going on, the real-time burden level measurement can be realized with high precision.

## 5. The Simulation and Industrial Verification Results

For the training of the network model, the hardware environment included Intel Xeon (R) w-2133CPU, NVIDIA Geforce, and one RTX2080 graphics card; the language we chose is python, and the platform we used is pytorch 1.7.1 and the CUDA version was 10.2. All our simulation diagrams were drawn based on MATLAB. To verify the correctness of the proposed methods of prediction and measurement of blast furnace burden level, the actual detection data of mechanical probe and radar probe in the same area of a steel plant blast furnace were collected for simulation. The radar probe sampled every 10 s, and the number of samples is 76,426 rows, where 66,426 rows of the data were used for model training and another 10,000 rows were used for model validation. The width of the sliding window was preferred as N=10. The data format is 76,424 × 10. The sampling interval of mechanical stock rod is 3∼6 min, and the number of samples is 3149.

### 5.1. Verification Results of the Blast Furnaces Burden Level Prediction and the Time Features Attraction Based on LFMN

LFMN is proposed to predict the blast furnace level in order to extract the time dimensional features of extreme time points such as speed, acceleration, jerk, and the change rate of jerk. For this purpose, it is required that the predicted burden level information can reflect the periodicity of burden level change and fit the real-time mechanical probe measurement data as much as possible, with a certain high accuracy. Figure 7 shows the trend of the prediction of burden under normal working conditions. It can be seen that the burden level data measured by the mechanical probe are mostly above and below the standard control stockline 1.5  m of the blast furnace, and the time of burden level data in a single cycle measured by the radar probe is kept within the 500  s left and right range. The reason for the radar data being much higher or lower than the mechanical probe data is that the radar uses microwaves to detect the burden level, and the microwaves are easily affected by the high-speed gas flow and dust in the blast furnace. However, due to the introduced guide layer mechanism and the slow variation features of the radar data, although the burden level predicted by LFMN has a certain deviation from the measured burden level data of the mechanical stock rod, it is highly consistent with the radar probe data in the change trend, especially in the fluctuation cycle of the data. This shows that the method of LFMN has high confidence in predicting blast furnace burden level information and extracting its time dimensional features for normal working conditions of the blast furnace.

Figure 8 shows that when the blast furnace is abnormal, the measured value of the mechanical probe usually deviates greatly from the standard control 1.5  m stockline of the burden, or even reaches the limit position 2.1  m, as shown in the figure, while the measured data of the radar probe often jumps greatly and frequently. From the prediction result of LFMN, under abnormal conditions, the accuracy of LFMN for burden level prediction decreases significantly for the reason that the network relies too much on radar data. In the case of an extremely bad blast furnace environment, the burden level changes periodically. The radar probe is affected by extreme environments, and the predicted burden level deviation is large. For example, at the time of 3600  s, the error between the predicted value of burden level and the measured value of the mechanical probe even exceeds 0.2  m. However, it can be found that due to abnormal furnace conditions, the blast furnace operator has significantly adjusted the blast furnace charging cycle into the period of 6030  s∼7030  s. The charging cycle in this period is much larger than 6 min in this cycle. After this cycle, the operator artificially shortened the blast furnace charging cycle to maintain the burden level at the standard control stockline. The figure shows that the length of charging cycle is roughly in the range of 300  s∼480  s after the time of 7210  s. At this time, the prediction method of LFMN can perfectly track the periodic change in the charging cycle and maintain considerable accuracy. In conclusion, it shows that the prediction method of LFMN proposed in this paper can not only effectively reflect the change trend of the burden level but also accurately predict both the periodicity and time change trend of the burden level under normal or abnormal conditions. In order to further verify the statistical significance and effectiveness of the proposed method of LFMN, a comparative statistical analysis on the consistency between the accuracy of the burden level predicted by the proposed method and the time period and the radar probe is conducted. The conclusions are shown in Table 1 below. From the statistical table, it can be found that the LFMN predicted error of the burden level is close to the radar probe measured error. The absolute error and relative error are 1.392% and 0.06, respectively. This shows that the method of LFMN can be used to directly predict the burden level of the blast furnace, which has a good reference value for the actual operation of the blast furnace. In addition, from the point of view of the coincidence degree of the blast furnace charging cycle, the method of LFMN reached the coincidence degree of 98.87%, which shows that the proposed method has high reliability in extracting time dimensional features of the blast furnace burden level information.

### 5.2. Verification Results of the Blast Furnace Burden Level Detection Based on ESST-RBFNN

ESST-RBFNN is proposed to realize real-time, accurate, continuous, and reliable detection of burden level information. Figure 9 shows the comparison of different methods for measuring burden levels under normal working conditions. According to the discrete burden level data measured by the mechanical probe, the burden level fluctuates around the standard 1.5  m stockline, and the length of the charging cycle is basically fixed. Although the radar data are continuous in real-time, it deviates from the accurate and reliable mechanical probe data at different time points to different degrees, indicating that its accuracy still needs to be greatly improved. The blast furnace burden level data predicted by traditional RBFNN can fit the periodicity and the change trend of the radar data to a certain extent, but it has a strong inhibition on the fluctuation of the burden level, meaning that the fluctuation range of the measured burden level is within 0.2  m, which obviously does not conform to the actual blast furnace charging site. At the same time, it cannot accurately track the changes in the mechanical probe data, and the overall detection level is high. This is because the traditional RBFNN does not take the correlation between the data measured by the mechanical probe and the radar probe into account. It only considers the radar data but ignores the accuracy and reliability of the mechanical probe data, which makes the accuracy and reliability of the detected data low and does not conform to the real production process. The ESST-RBFNN extracts accurate time dimensional features and space dimensional features of the burden level through the method of LFMN and the space feature regression model based on radar data. It also deeply excavates and utilizes the internal relationship between the two detection data. The detected burden level data can not only perfectly fit the periodicity and change trend of radar probe data, but also track the changes in mechanical probe data in real-time with high precision. It greatly improves the accuracy and reliability of the detection data, which shows that the burden level data detected by this method not only has the real-time and continuous advantages of radar data but also has the accurate and reliable advantages of mechanical probe data. This shows its high practical value.

Figure 10 shows the comparison between the radar data, the detection data of the traditional RBFNN and the ESST-RBFNN under abnormal working conditions. Under this abnormal working condition, the mechanical probe data jumps up and down greatly. It can be seen from Figure 10 that at this time, the radar data can hardly track the mechanical probe data, and its accuracy is very low. This shows that under abnormal working conditions, due to a large amount of dust in the blast furnace, the reference value of the radar data is limited. The traditional RBFNN also has the disadvantage of obviously restraining the fluctuation of burden level, and its sensitivity is poor. At the same time, similar to the radar data, the detection data of the traditional RBFNN cannot track the mechanical probe data, so its reliability and accuracy are not high. The ESST-RBFNN proposed in this paper is not only perfectly consistent with the periodicity and change trend of the radar data but can also accurately track the mechanical probe data most of the time. Only in the case of extremely accidental anomalies, such as the three outliers shown in the figure, when the burden level suddenly jumps sharply and changes greatly, the tracking effect at the outlier position is poor. However, compared with the radar data, the reliability has been significantly improved.

Figure 11 and Figure 12 show the absolute error and relative error distributions of the burden level data measured by radar and the methods of traditional RBFNN and ESST-RBFNN compared with the measured data of mechanical stock rod. The area between the black dotted lines represents the main distribution area of the data measured by ESST-RBFNN. It can be seen from the figure that the absolute error and relative error of radar data fluctuate greatly. The absolute error is within the range of ±0.5  m, and the relative error even reaches 40%. The absolute error and relative error of the traditional RBFNN method are significantly improved compared with the radar probe; however, the conditions of burden level lower than 1.4  m are minimal during the actual operation of the blast furnace, and the traditional RBFNN method relies on the sample data too much, which leads to the error distribution points of the method concentrated in the upper area of the figure, that is, the measured value has an obvious larger error compared with the real value. Therefore, the maximum relative error of the traditional RBFNN method reaches up to 25%. For the actual blast furnace charging process controlled by the standard 1.5  m stockline, the errors of the above two methods for burden level measurement are unacceptable in engineering, and the practical application value of which is thus low. The proposed ESST-RBFNN overcomes the problem that the traditional RBFNN is excessively dependent on the data samples by using the space-time features of burden level distribution. The absolute error and relative error range shown in the figure are within the range highlighted by the red line. The absolute error is within the range of ±0.1  m and the relative error is within the range of ±5%. At the same time, considering the relative error of the mechanical probe data itself, on the premise of realizing the real-time continuous measurement of the blast furnace burden level, the measured error of the burden level detection method proposed in this paper is very close to the measured error of the mechanical probe, which satisfies the requirements of the blast furnace smelting process and has very high practicality.

Figure 13 shows the 45° line diagram of the measurement effect of the above three methods to reflect the accuracy of the three material level measurement methods more intuitively. It can be seen from the figure that the detection data obtained by the method of ESST-RBFNN are distributed in the red measured standard line, forming a vertebral body, whose accuracy is significantly high. However, compared with the detection methods of radar probe and traditional RBFNN, the method of ESST-RBFNN has the closest detection data to the standard line with the highest accuracy. When the standard line is within 1.5∼2.0  m, the data of three measurement methods are more distributed. The radar data deviate far from the standard line with low accuracy. The traditional RBFNN measurement data also deviate relatively far from the standard line and have the inherent error of high detection level, and its accuracy is also low. The detection data of the ESST-RBFNN method are concentrated and closely distributed around the standard line, which can track the standard burden level well, with high accuracy and reliability. When the standard line is higher than 2.0  m, it indicates that the blast furnace has an abnormal working condition. From the figure, it indicates that the detection methods of a radar probe and traditional RBFNN are far from the standard line, and the accuracies of which are much lower. The method of ESST-RBFNN has little fluctuations and can still track the measured burden level with high precision.

To further verify the correctness and effectiveness of the proposed method, the 24-h operation data based on the 10,000 rows of validation data of the ESST-RBFNN method were collected to draw the operation effect diagram, as shown in Figure 14. The absolute error, relative error and hit rate of radar probe, and the methods of traditional RBFNN and ESST-RBFNN for up to 3 months are counted, as shown in Table 2. It can be seen from Figure 14 that based on the 10,000 rows of validation data within 24 h of operation, whether under normal or abnormal conditions, the detection method of ESST-RBFNN can well track the mechanical probe data with high accuracy and reliability. At the same time, it can well fit the periodicity and variation trend of the radar data and has good real-time continuity. From the 3-month statistical error analysis in Table 2, the MRE (Mean Relative Error) and RMSE (Root Mean Square Error) of the burden level detection method of ESST-RBFNN are nearly four times higher than those of the radar probe and the method of traditional RBFNN. The hit rate of samples with relative error lower than 2% reaches up to 91.17%. The hit rate of samples with relative error lower than 5% even reaches up to 99.33%. Compared with the radar probe and the method of traditional RBFNN, there is a significant qualitative improvement. The statistical analysis data further illustrate that the burden level detection method of ESST-RBFNN proposed in this paper can obtain continuous and accurate blast furnace burden level information with high precision.

## 6. Conclusions and Discussion

### 6.1. Conclusions

Real-time and continuous blast furnace burden level detection with high accuracy and high reliability is of great significance to realizing green and fine production of blast furnaces. The traditional mechanical probe detection method and radar detection method cannot obtain real-time accurate and continuous burden level information. The proposed method combines the advantages of high precision and reliability of mechanical probe data and the real-time continuous characteristics of radar data. Based on the slow variation features of burden level change in the time dimension and fast variation features in the space dimension, it can obtain real-time accurate and continuous burden level information with high precision and great reliability.

Based on the slow variation features of radar data, the LFMN network can predict the blast furnace burden level with relatively high accuracy and extract time dimensional features of the burden level information with great reliability. Combined with the proposed ESC algorithm, the ESST-RBFNN can realize a self-renewing dynamic structure with the ability of online training and provide a better approximation performance than the traditional RBFNN. The simulation results and industrial verification indicate that the proposed method can realize real-time and continuous blast furnace burden level detection with high accuracy, high stability, and great reliability. Therefore, the proposed method has a high industrial application value.

### 6.2. Discussion

The proposed method can track the burden level information well under normal working conditions and most abnormal working conditions, which obtains real-time and continuous burden level information with high precision and great reliability. However, under some extremely abnormal working conditions, when some extremely large fluctuations occur, the accuracy and reliability of burden level detection will decrease to some extent, where its practical value can be improved by further optimization. The main reason is that the LFMN network relies too much on the slow variation features of radar data, which means that when the burden level fluctuates significantly, the prediction of burden level based on the slow variation features of radar data will not track the real information of the burden level change trend, which needs to be further studied and improved in the future.

## Figures and Tables

**Figure 1 sensors-22-05412-f001:**
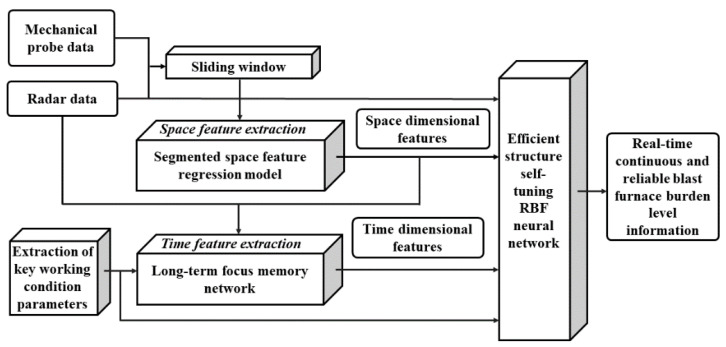
Research framework of the blast furnace burden level prediction and detection.

**Figure 2 sensors-22-05412-f002:**
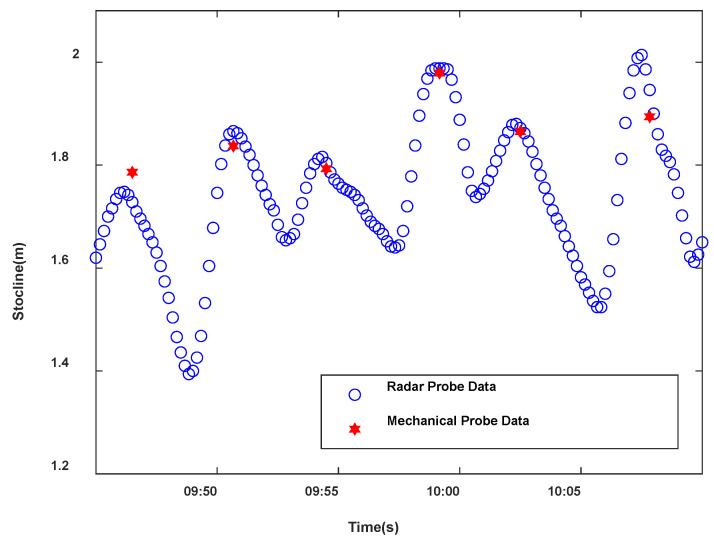
Detection data of radar and mechanical probe.

**Figure 3 sensors-22-05412-f003:**
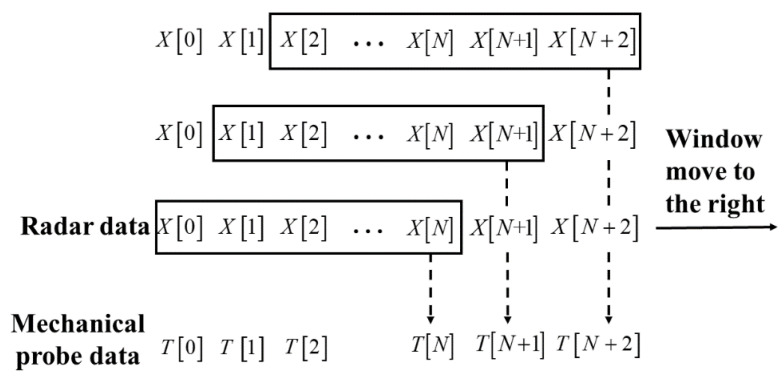
Schematic of the data processing process of the sliding window.

**Figure 4 sensors-22-05412-f004:**
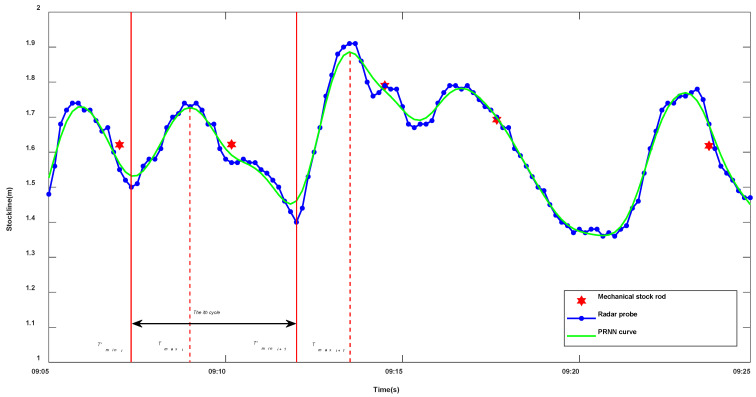
Space feature regression model curve of radar data.

**Figure 5 sensors-22-05412-f005:**
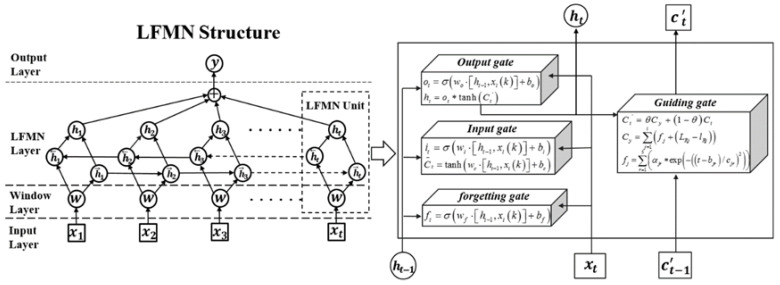
Structure of the long-term focus memory network.

**Figure 6 sensors-22-05412-f006:**
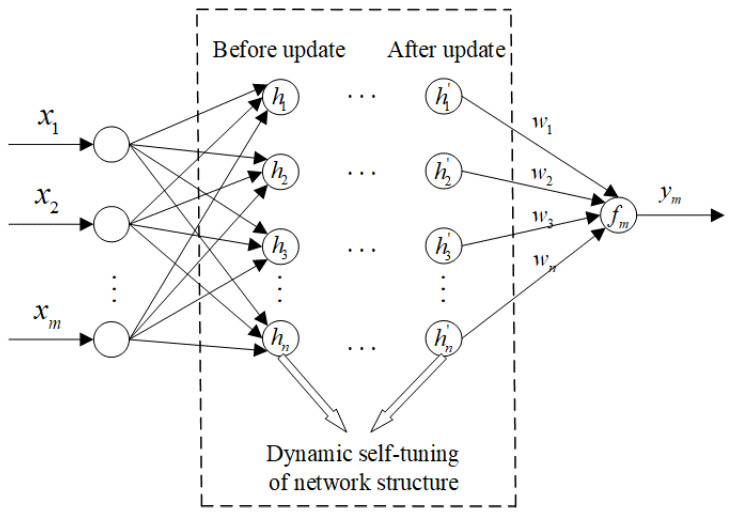
Network structure of ESST-RBFNN.

**Figure 7 sensors-22-05412-f007:**
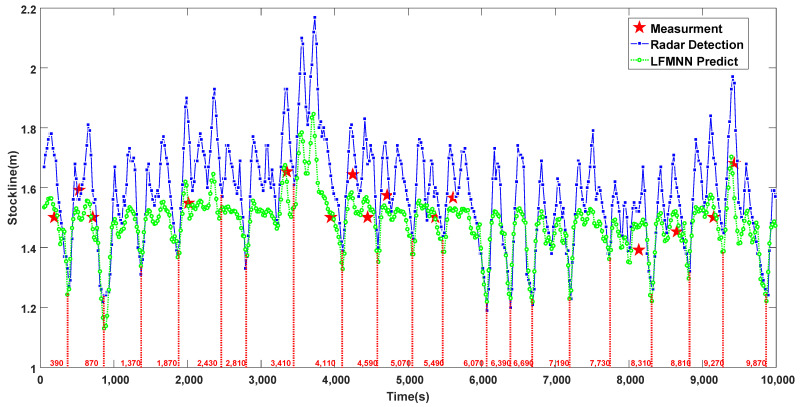
Trend of burden level prediction under normal working conditions.

**Figure 8 sensors-22-05412-f008:**
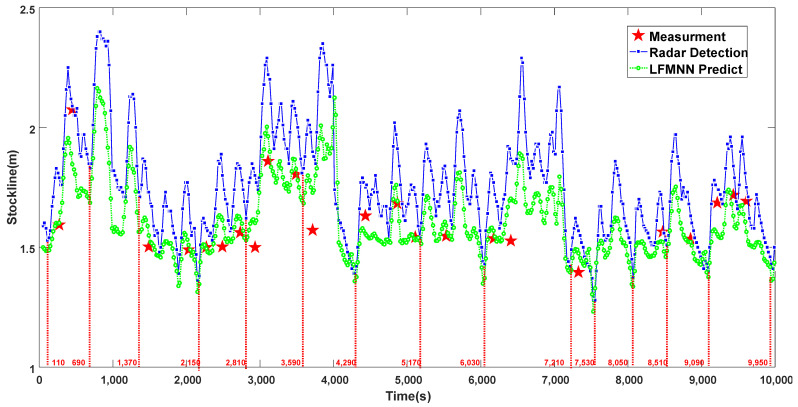
Trend of burden level prediction under abnormal working conditions.

**Figure 9 sensors-22-05412-f009:**
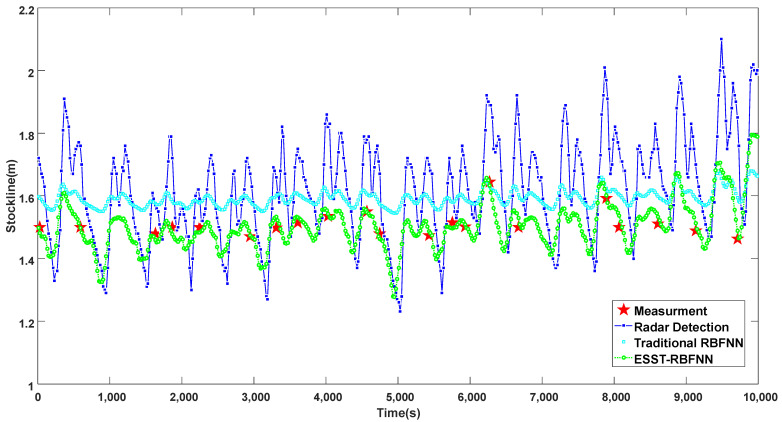
Comparison chart of different methods for measuring blast furnace burden level under normal working conditions.

**Figure 10 sensors-22-05412-f010:**
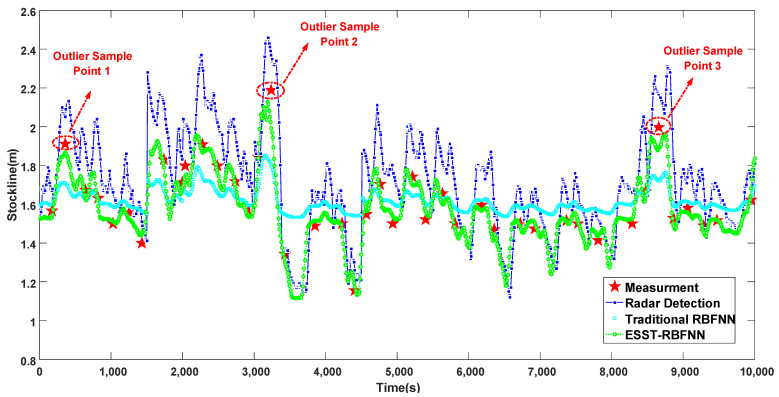
Comparison chart of different methods for measuring blast furnace burden level under abnormal working conditions.

**Figure 11 sensors-22-05412-f011:**
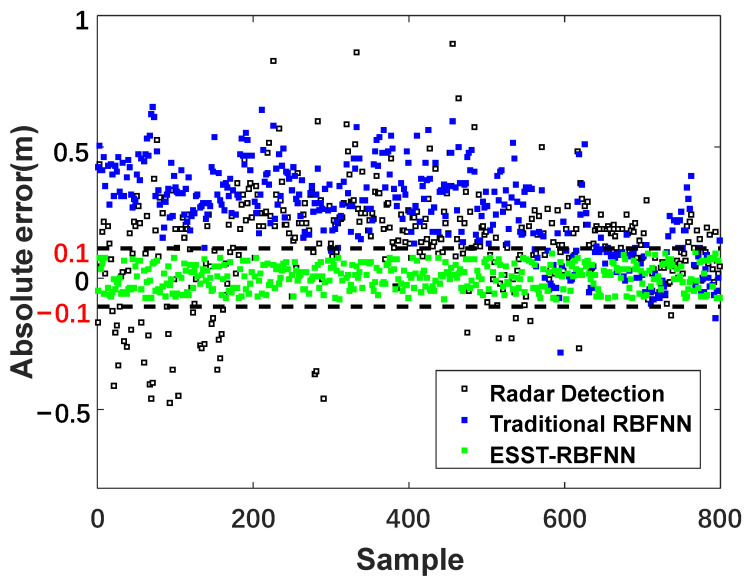
Compassion diagram of absolute error.

**Figure 12 sensors-22-05412-f012:**
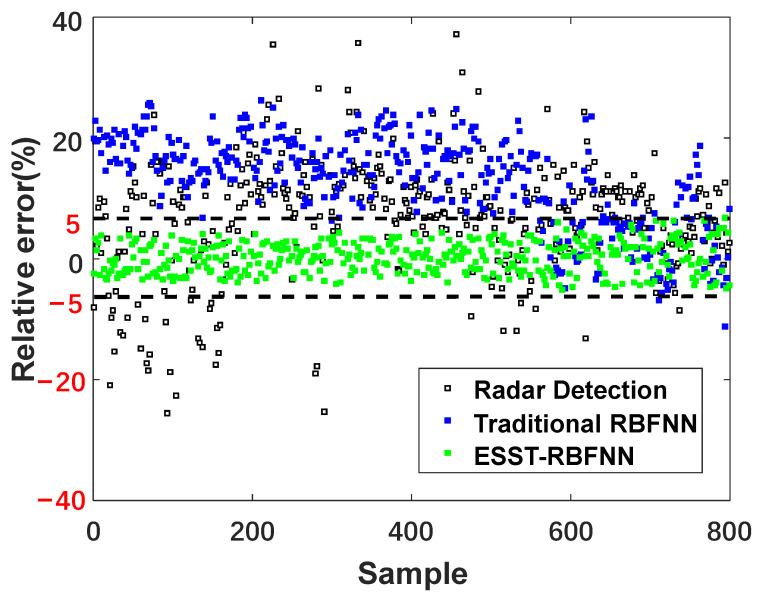
Compassion diagram of relative error.

**Figure 13 sensors-22-05412-f013:**
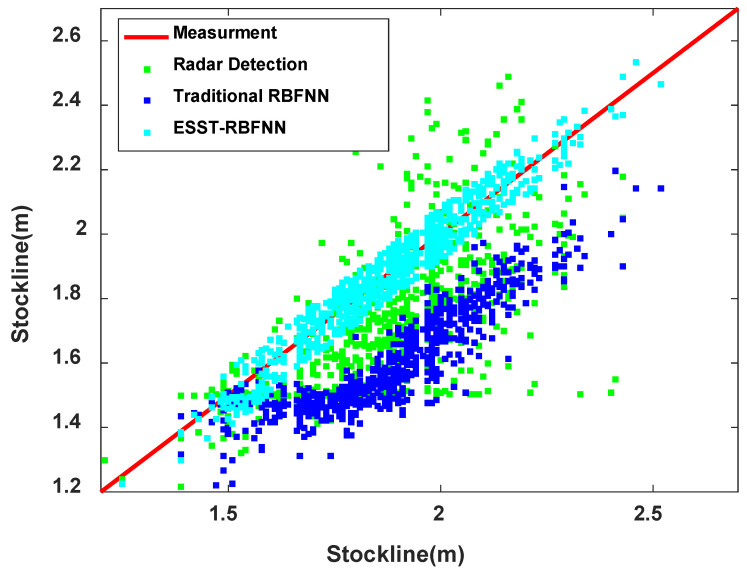
The 45° line diagram of the measurement effect of the three methods.

**Figure 14 sensors-22-05412-f014:**
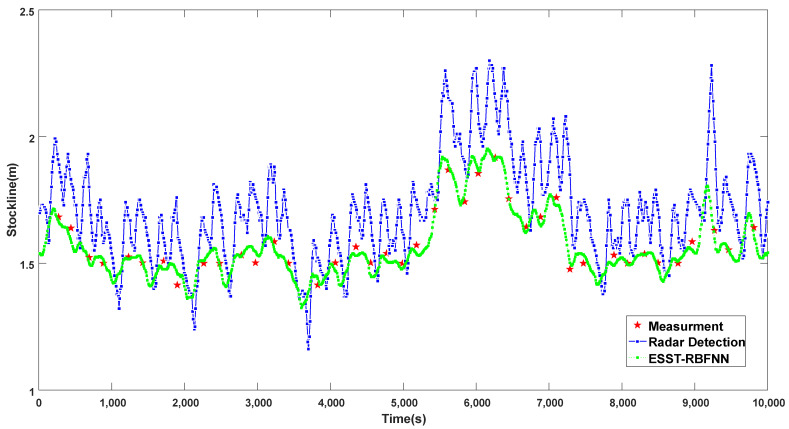
Operation effect diagram of the detection method of ESST-RBFNN based on the validation data.

**Table 1 sensors-22-05412-t001:** Statistical prediction error comparison.

Method	Statistical Indices		
	MRE	RMSE	Cycle-Fit
Radar Probe	7.182%	0.176	99.53%
LFMN	8.574%	0.236	98.87%

**Table 2 sensors-22-05412-t002:** Statistical error comparison.

Method	Statistical Indices			
	MRE	RMSE	Error-2%	Error-5%
Radar	8.573%	0.1847	13.97%	32.53%
RBFNN	7.291%	0.1825	28.91%	66.50%
Proposed	2.361%	0.0480	91.17%	99.33%

## Data Availability

Data available on request due to privacy. The data presented in this study are available on request from the corresponding author. The data are not publicly available for the reason that it is real-time industrial operation data. In order to be responsible for the industry, it is difficult for us to disclose these data, but you can obtain them from the corresponding author by email.

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
