# Peer review of "Real-Time Detection and Short-Term Prediction of Blast Furnace Burden Level Based on Space-Time Fusion Features"

_sensors, 2022, doi:10.3390/s22145412_

Round 1
Reviewer 2 Report
This article proposes a new blast furnace burden level detection method to obtain real-time, continuous, accurate and reliable blast furnace burden level information. I think the interesting part of this article is to take the prediction of burden level as a significant step of the detection of burden level. This paper proposes a new network LFMN for blast furnace burden level prediction, which can predict the blast furnace burden level in a short term. Its purpose is to extract the time characteristics of blast furnace burden level, and then it can be used for blast furnace burden level detection. Its simulation results also show that the effect of long-term detection through short-term prediction is great. At the same time, this article constructs the ESST-RBFNN network, which can detect the blast furnace burden level and obtain real-time continuous burden level information with high precision and accuracy. To the end, I think the whole article has high innovation and industrial application value.
To further extend, the following few suggestions may help to improve the quality of the article:
1) When using the sliding window to construct the single shot relationship between the data of radar probe and mechanical probe, it is not mentioned how to select the size of sliding window . maybe there can be a simple explanation。
2)In this article, the fifth order polynomial is used to carry out piecewise nonlinear fitting of radar data. How to determine the fitting order may be briefly explained.
3)There are a few spelling mistakes in the article, as an example, in Figure 4, “Mechanical stock rod” is misspelled as “Mechanical stack rob”.
4)The name “PRNN” does not appear in the text, but directly appears in Figure 4 for the first time. Maybe it should be explained in the text first.
5)For the composition description of the sample input in Equation 3, and may be written wrong accidentally, they may be and .
6)Equation 23 shows the lower bound of the optimization objective is . However, according to the derivation of equation 22, the lower bound of the optimization objective should be .
Reviewer 3 Report
This study numerically develops a LFMN-ESST-RBFNN approach for the prediction of the burden level of blast furnaces, which is quite interesting in terms of the concept. The work provides information for a better understanding of ironmaking. It is recommended for minor revision after addressing the following points:
1. What are the language and software or platform selected in this study?
2. Line 106, page 3, what is the definition of the “next cycle” that is to be predicted and why? What factors will affect this cycle? For example, will it be different if the burden materials change or in another blast furnace?
3. Line 2, page 6, it is recommended to add an introduction of the advantage and disadvantages of the selected “LFMN deep network structure” to the introduction or just here. In other words, why it is selected based on the slow variation feature of radar data?
4. Lines 1-5 of Section 4, page 8, similarly, why “ESST-RBFNN” is selected based on the slow change features in the time dimension and fast change features in the space dimension of the burden level?
5. Lines 1-6 of Section 5, page 12, I would like to double confirm that the database of radar probe samples for training and testing is 76426 rows. If so, it is recommended to add a database of another cycle that is not within the 76424 rows for a demonstration of the model prediction.
6. Figure 7 and Figure 8, it shows that the radar-detected data are largely higher than measurement data under normal/abnormal working conditions, is it still ok to use radar-detected data to train the model? Also, most of the LFMNN predicted data are also smaller than Radar detected data.
7. Please add a nomenclature.
8. Please add the conclusions.
